# Deficiency of IQCH causes male infertility in humans and mice

**Tiechao Ruan[1,2†], Ruixi Zhou[1,2†], Yihong Yang[3†], Junchen Guo[4,5], Chuan Jiang[1], Xiang Wang[1], Gan Shen[1], Siyu Dai[1], Suren Chen[6]\*, Ying Shen[1,7]\***

[1]Key Laboratory of Obstetrics, Gynecologic and Pediatric Diseases and Birth Defects of the Ministry of Education, West China Second University Hospital, Sichuan University, Chengdu, China; [2]Department of Pediatrics, West China Second University Hospital, Sichuan University, Chengdu, China; [3]Reproduction Medical Center of West China Second University Hospital, Key Laboratory of Obstetric, Gynecologic and Pediatric Diseases and Birth Defects of Ministry of Education, Sichuan University, Chengdu, China; [4]Sichuan University-The Chinese University of Hong Kong (SCU-CUHK) Joint Laboratory for Reproductive Medicine, Key Laboratory of Obstetric, Gynaecologic and Paediatric Diseases and Birth Defects of Ministry of Education, West China Second University Hospital, Sichuan University, Chengdu, China; [5]Reproductive Endocrinology and Regulation Laboratory, Department of Obstetric and Gynaecologic, West China Second University Hospital, Sichuan University, Chengdu, China; [6]Education Key Laboratory of Cell Proliferation & Regulation Biology, College of Life Sciences, Beijing Normal University, Beijing, China; [7]NHC Key Laboratory of Chronobiology, Sichuan University, Chengdu, China

**\*For correspondence:**
chensr@bnu.edu.cn (SC);
yingcaishen01@163.com (YS)

[†]These authors contributed equally to this work

**Competing interest:** The authors declare that no competing interests exist.

**Abstract** IQ motif-containing proteins can be recognized by calmodulin (CaM) and are essential for many biological processes. However, the role of IQ motif-containing proteins in spermatogenesis is largely unknown. In this study, we identified a loss-of-function mutation in the novel gene IQ motif-containing H (*IQCH*) in a Chinese family with male infertility characterized by a cracked flagellar axoneme and abnormal mitochondrial structure. To verify the function of IQCH, *Iqch* knockout (KO) mice were generated via CRISPR-Cas9 technology. As expected, the *Iqch* KO male mice exhibited impaired fertility, which was related to deficient acrosome activity and abnormal structures of the axoneme and mitochondria, mirroring the patient phenotypes. Mechanistically, IQCH can bind to CaM and subsequently regulate the expression of RNA-binding proteins (especially HNRPAB), which are indispensable for spermatogenesis. Overall, this study revealed the function of IQCH, expanded the role of IQ motif-containing proteins in reproductive processes, and provided important guidance for genetic counseling and genetic diagnosis of male infertility.

## eLife assessment

This **valuable** study describes mice with a knock out of the IQ motif-containing H (IQCH) gene, to model a human loss-of-function mutation in IQCH associated with male sterility. While the evidence for interaction between IQCH and potential RNA binding proteins is limited, the human infertility is reproduced in the mouse, making it a **compelling** model. The paper could be of interest to cell biologists and male reproductive biologists working on the sperm flagellar cytoskeleton and mitochondrial structure.

## Introduction

Spermatogenesis is one of the most complex biological processes in male organisms and functions to produce mature spermatozoa from spermatogonia in three phases: (i) mitosis, (ii) meiosis, and (iii) spermiogenesis (*Hess and Renato de Franca, 2008*). This delicate process can be easily disturbed and further cause male reproductive disorders. Male infertility affects 7% of men in the general population, and its causes vary and include anatomical or genetic abnormalities, systemic diseases, infections, trauma, iatrogenic injury, and gonadotoxins (*Krausz et al., 2018a*). Approximately 10% of the human genome is related to reproductive processes; thus, male infertility is often predicted to be largely genetic in origin, whereas only 4% of infertile men are diagnosed with a distinct genetic cause (*Krausz and Riera-Escamilla, 2018b*). Genetic causes are highly heterogeneous and involve chromosomal abnormalities, point mutations in single genes, copy number variations, microRNA dysfunction, and polygenic defects (*Meschede and Horst, 1997*; *Traven et al., 2017*). The highest percentage of known genetic factors that account for up to 25% of male infertility are karyotype anomalies, Y chromosome microdeletions, and *CFTR* mutations, which are mostly associated with azoospermia (*Krausz and Riera-Escamilla, 2018b*). However, in a relatively high proportion of infertile men (40%), the etiology cannot be recognized and is also referred to as idiopathic. With the help of assisted reproductive techniques (ART), some men have the chance to reproduce; however, there is a risk of passing on undetermined genetic abnormalities. In addition, genetic defects leading to fertilization failure and embryo development arrest cannot be effectively rescued by ART; thus, discovering novel genetic factors and further confirming their molecular mechanisms are of clinical importance.

CaM is defined as a major $Ca^{2+}$ sensor that activates many kinds of enzymes in response to an increase in intracellular $Ca^{2+}$ through interactions with a diverse group of cellular proteins (*Klee et al., 1980*; *Means et al., 1982*). CaM has a completely conserved amino acid sequence across all vertebrates (*Jensen et al., 2018*). Specifically, in the human genome, there are three independent CaM genes (*CALM1-3*), while in the mouse genome, there are five CaM genes (*Calm1-5*) (*Jensen et al., 2018*; https://www.ncbi.nlm.nih.gov/gene/?term=calm+mouse). Previous studies have reported that CaM is extraordinarily abundant in the mammalian testis and brain (*Smoake et al., 1974*; *Kakiuchi et al., 1982*). Moreover, CaM is predominantly expressed in the mammalian testis, from the pachytene to meiotic division stages (*Smoake et al., 1974*; *Kakiuchi et al., 1982*; *Yamamoto, 1985*; *Sano et al., 1987*; *Kägi et al., 1988*; *Moriya et al., 1995*). CaM binds to proteins through recognition motifs, including the short CaM-binding motif containing conserved Ile and Gln residues (IQ motif) for $Ca^{2+}$-independent binding and two related motifs for $Ca^{2+}$-dependent binding (*Nie et al., 2009*). The identified IQ motif-containing proteins included a good range of proteins with biological functions, one of which is a sperm surface protein, suggesting that IQ motif-containing proteins might play a potential role in male reproductive processes (*Wen et al., 1999*; *Dolmetsch et al., 2001*). Currently, 15 IQ motif-containing proteins have been identified in humans, namely, IQCA1, IQCB1, IQCC, IQCD, IQCE, IQCF1, IQCF2, IQCF5, IQCF6, IQCG, IQCH, IQCJ, IQCK, IQCM, and IQCN. With the exception of IQCF2, mice possess a similar set of IQ proteins (https://www.ncbi.nlm.nih.gov/gene/?term=IQCF2). Notably, IQCF4 is present in mice but absent in humans. However, among these IQ motif-containing proteins, only a few have been reported to be responsible for male fertility. It has been shown that the IQ motif-containing D (IQCD) primarily accumulates in the acrosome area of spermatids during mouse spermiogenesis, and the acrosome reaction is inhibited in human spermatozoa by the anti-IQCD antibody, suggesting a potential function of IQCD in fertilization and the acrosome reaction (*Zhang et al., 2019*). Moreover, *Iqcf1* KO male mice showed significantly less fertility, which was related to reduced sperm motility and acrosome reactions (*Fang et al., 2015*). Importantly, *Iqcg* is required for mouse spermiogenesis, which is attributable to its role in flagellar formation and/or function (*Harris et al., 2014*). Humans and mice without IQCN presented failed fertilization rates related to manchette assembly defects (*Dai et al., 2022*). Although reliable findings provide substantial insight into IQ motif-containing proteins that participate in male reproductive processes, the relevant regulatory mechanism and the function of many other IQ motif-containing proteins in spermatogenesis have not been determined thus far.

Here, we revealed a testis-specific IQ motif containing the H gene, *IQCH*, which is essential for spermiogenesis and fertilization. Disruption of IQCH leads to deficient acrosome activity and abnormal structure of the axoneme and mitochondria in both humans and mice. Moreover, it is suggested that the interaction of IQCH with CaM is a prerequisite for IQCH function, which further regulates the

expression of RNA-binding proteins (especially HNRPAB) during spermatogenesis. Collectively, our findings unveiled a novel genetic diagnostic indicator of male infertility, and the uncovered mechanism of IQCH in spermatogenesis might shed new light on the treatment of this disease.

## Results

### Identification of a novel homozygous variant of *IQCH* involved in male infertility

A 33-year-old man from a consanguineous family with primary infertility for four years was recruited for our study (*Figure 1A*). Semen analysis of this patient revealed inordinate decreases in sperm motility and count, as well as abnormal sperm morphology (*Table 1*). In addition to the proband, his great uncle was also affected, and he had never conceived during sexual intercourse without contraception in his marriage. We further explored the possible genetic cause through whole-exome sequencing (WES) analysis, and a homozygous deletion variant in *IQCH* (c.387+1_387+10 del, NM_0010317152) was identified in the proband. This variant is rare in the general human population according to the 1000 Genomes Project (0.007%), ExAC Browser (0.059%), and gnomAD databases (0.024%). We further clarified the putative contribution of this variant in this family by Sanger sequencing. Noticeably, this homozygous variant was confirmed in his infertile great uncle, and the fertile parents of the proband carried the heterozygote of this variant (*Figure 1A*).

Moreover, we used a minigene splicing assay to examine the effect of this variant on *IQCH* mRNA splicing. The electrophoresis results revealed that wild-type *IQCH* (WT-*IQCH*) yielded one transcript with a size of 381 bp, whereas the mutant *IQCH* (Mut-*IQCH*) resulted in one strong band of 263 bp (*Figure 1B and i*). Sanger sequencing further showed that the variant resulted in the absence of the whole exon 4, which was expected to translate into a truncated protein (*Figure 1B, ii and iii*). Therefore, we constructed a plasmid containing the aberrant cDNA sequence of *IQCH* caused by the splicing mutation to verify the effect of the variant on protein expression. The western blotting results showed that the Mut-c*IQCH* plasmid did not express IQCH, while the WT-c*IQCH* plasmid did (*Figure 1B, iv*). We further conducted immunofluorescence staining of spermatozoa from the proband and the control. We detected very little expression of the IQCH protein in spermatozoa from the proband compared to the control (*Figure 1C*). Taken together, these results reveal that this identified *IQCH* variant is responsible for splicing abnormalities and further causes a lack of IQCH expression, which is likely the genetic cause of male infertility in this family.

### IQCH deficiency leads to sperm with cracked axoneme structures accompanied by defects in the acrosome and mitochondria

A comprehensive sperm morphology analysis was further conducted on the proband (the semen was not available from his great uncle). Papanicolaou staining revealed that the spermatozoa displayed multiple morphological flagellar abnormalities, such as coiled, bent, irregular, and even fractured tails (*Figure 2A*). These morphological anomalies were confirmed more precisely by scanning electron microscopy (SEM) analysis (*Figure 2B*). Significantly, more subtle abnormalities, such as axoneme exposure, bending, and cracking, were further identified between the midpiece and the principal piece of the proband's spermatozoa (*Figure 2B*). As expected, a deranged or incomplete '9+2' microtubule structure of the flagella was observed in the proband by transmission electron microscopy (TEM) analysis (*Figure 2—figure supplement 1*). In addition, we detected ultrastructural defects in the spermatozoa nucleus, including irregular shapes, large vacuoles, and deformed acrosomes (*Figure 2C*). The mitochondria of the spermatozoa had an abnormal arrangement and enlarged diameter (*Figure 2C*).

Moreover, we performed immunofluorescence staining for the marker of the acrosome (peanut agglutinin: PNA) as well as the mitochondrial marker (transcription factor A, mitochondrial: TFAM) to confirm the deficiency of the acrosomes and mitochondria in the proband's spermatozoa. The results of the PNA and TFAM staining suggested that the spermatozoa acrosomes and mitochondria were severely defective in the proband compared to the control (*Figure 2D and E*). In addition, we performed staining for SEPTIN4 (SEPT4), a functional marker of the annulus, to explore whether the flagellar fracture at the joint between the middle piece and the principal piece was caused by a nonfunctional annulus. There was no significant difference in the SEPT4 signal in the spermatozoa between the proband and the control (*Figure 2—figure supplement 1*).

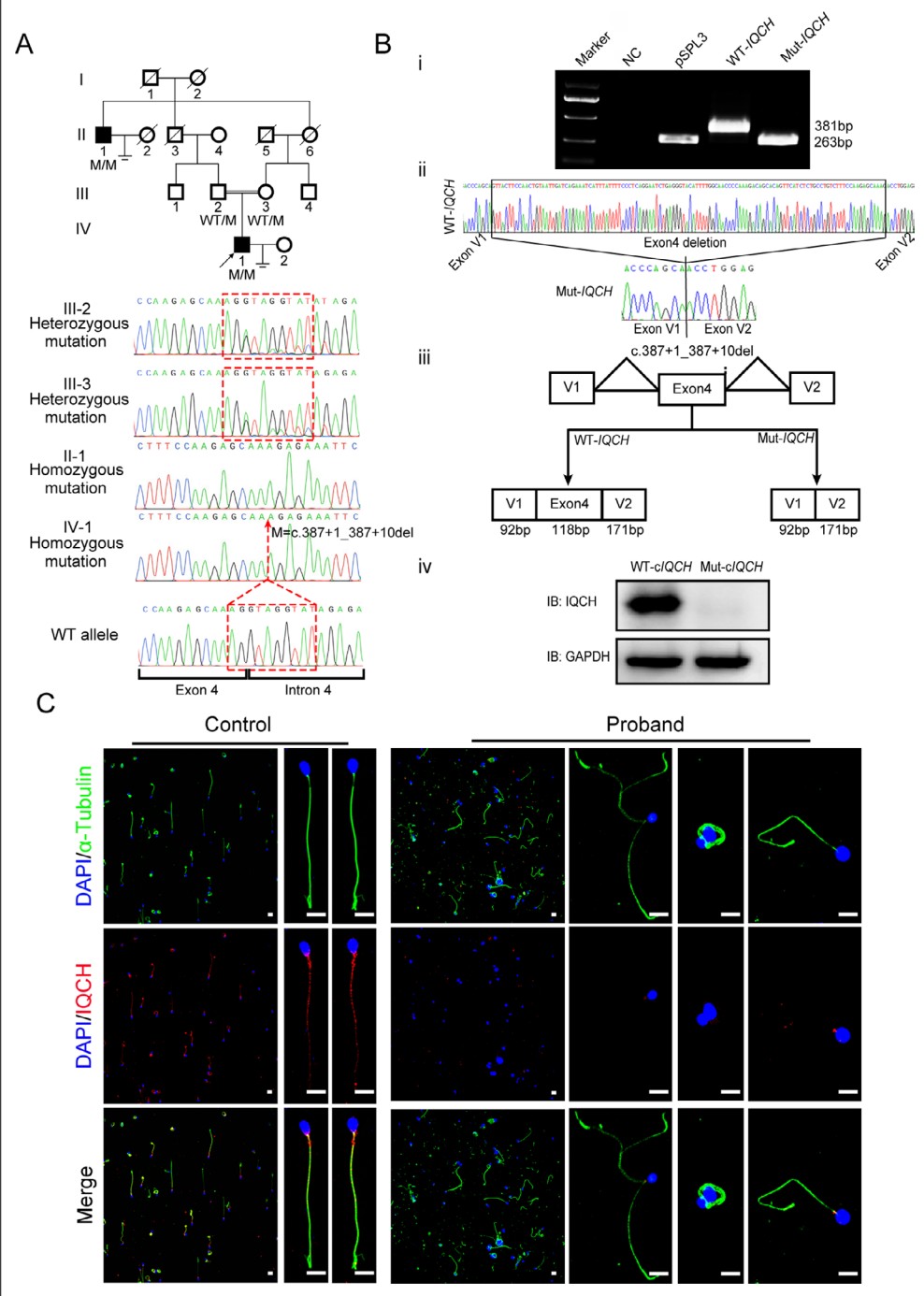

**Figure 1.** Identification of a homozygous splicing mutation in IQ motif-containing H (*IQCH*) in a consanguineous family with male infertility. (**A**) Pedigree analysis of the consanguineous family with two infertile males (II-1 and IV-1), with the black arrow pointing to the proband. Sanger sequencing revealed that the affected males exhibited a homozygous variant in *IQCH*. Sequence chromatograms are shown below the pedigree. (**B**) (**i**) The electrophoresis results of the minigene assay show a decrease in the molecular weight of the RT-PCR products generated from Mut-*IQCH* (263 bp) compared with those from WT-*IQCH* (381 bp). (**ii**) Sanger sequencing of the complementary DNA of the splicing mutation showing the deletion of exon 4 in *IQCH*. (**iii**) The pattern diagram demonstrating the splicing effects caused by the *IQCH* mutation. (**iv**) Western blotting results showed that the Mut–c*IQCH* plasmids did not express IQCH. NC, negative control. Three independent experiments were performed. (**C**) Immunofluorescence staining showed that the expression of IQCH was barely detected in the proband's sperm compared with that in the control (blue, DAPI; green, α-Tubulin; red, IQCH; scale bars, 5 μm).

*Figure 1 continued on next page*

*Figure 1 continued*

The online version of this article includes the following source data for figure 1:

**Source data 1.** Primers for Sanger sequencing and Minigene.

**Source data 2.** Original blots and gels of *Figure 1*.

## Impairment of male fertility in mice without *Iqch*

To further elucidate the role of IQCH in male reproduction, we first explored the pattern of *Iqch* expression in mice by quantitative real-time PCR (qPCR) and found that *Iqch* was more highly expressed in the mouse testis than in other organs (*Figure 3—figure supplement 1*). Additionally, we investigated the temporal expression of *Iqch* in mouse testes on different postnatal days. The results revealed that the expression of *Iqch* significantly increased on postnatal day 21, peaked on postnatal day 35, and then stabilized (*Figure 3—figure supplement 1*). To better understand the role of IQCH in spermatogenesis, we performed immunofluorescence staining of germ cells at different developmental stages in human and mouse testes. IQCHs exhibited the same pattern in human and mouse spermatogenesis and were detected mainly in the cytoplasm of spermatocytes and round spermatids and in the flagella of late spermatids (*Figure 3—figure supplement 1*). The testis-enriched expression of *Iqch*/IQCH suggested its potential role in spermatogenesis.

We next generated *Iqch* KO mice using clustered regularly interspaced short palindromic repeat (CRISPR)-Cas9 technology. We used a guide RNA targeting exons 2 through 3 of *Iqch* to achieve the knockout (*Figure 3—figure supplement 2*). Polymerase chain reaction (PCR), reverse transcription-PCR (RT–PCR), and western blotting further confirmed the success of the *Iqch* KO mouse construction (*Figure 3—figure supplement 2*). The *Iqch* KO mice showed no overt abnormalities in their development or behavior. The *Iqch* KO female mice showed normal fertility, and hematoxylin-eosin (H&E) staining further showed that the *Iqch* KO female mice had normal follicle development compared to that of the WT mice (*Figure 3—figure supplement 2*). However, the fertility of the KO male mice was significantly lower than that of the WT mice, as was the pregnancy rate and litter size (*Figure 3—figure supplement 2*). There was no detectable difference in the testis or epididymis/body weight ratio between the WT and *Iqch* KO mice (*Figure 3—figure supplement 2*), and the histology of the testes and epididymides in the *Iqch* KO mice showed no obvious abnormalities compared to those in the WT mice (*Figure 3—figure supplement 3*).

However, computer-assisted sperm analysis (CASA) confirmed that sperm motility was significantly reduced in the KO mice, and the sperm count was slightly decreased (*Table 2*; *Figure 3—video 1* and *Figure 3—video 2*). Moreover, morphological anomalies of the sperm flagella, such as an unmasking, bending, or cracking axoneme, which recapitulated the flagellar phenotype of the infertile patient, were easily observed (*Figure 3A and B*). We further found that during the spermatogenic process, abnormalities in axoneme exposure had already occurred when the spermatozoa flagellum developed (*Figure 3—figure supplement 4*). Intriguingly, tail defects, including bending and cracking between the middle piece and the principal piece, were mainly present in the epididymal spermatozoa (*Figure 3—figure supplement 4*), suggesting that more severe flagellum breakage might occur during sperm movement.

We further performed TEM analysis to investigate the ultrastructural defects in the testicular and epididymal spermatozoa of the *Iqch* KO mice. In contrast to those of the WT mice, the spermatozoa

**Table 1.** Results of the semen analysis and sperm morphology examination of the patient.

| Parameter | Patient | Reference* |
|---|---|---|
| Sperm volume, ml | 2.8 | ≥1.5 |
| Sperm concentration, million/ml | 10.0 | ≥15 |
| Vitality, % | 30 | ≥58 |
| Motility, % | 4 | ≥32 |
| Abnormal morphology (%) | 99.5 | - |

*The reference values were based on the guidelines provided in the WHO manual fifth edition (2010).

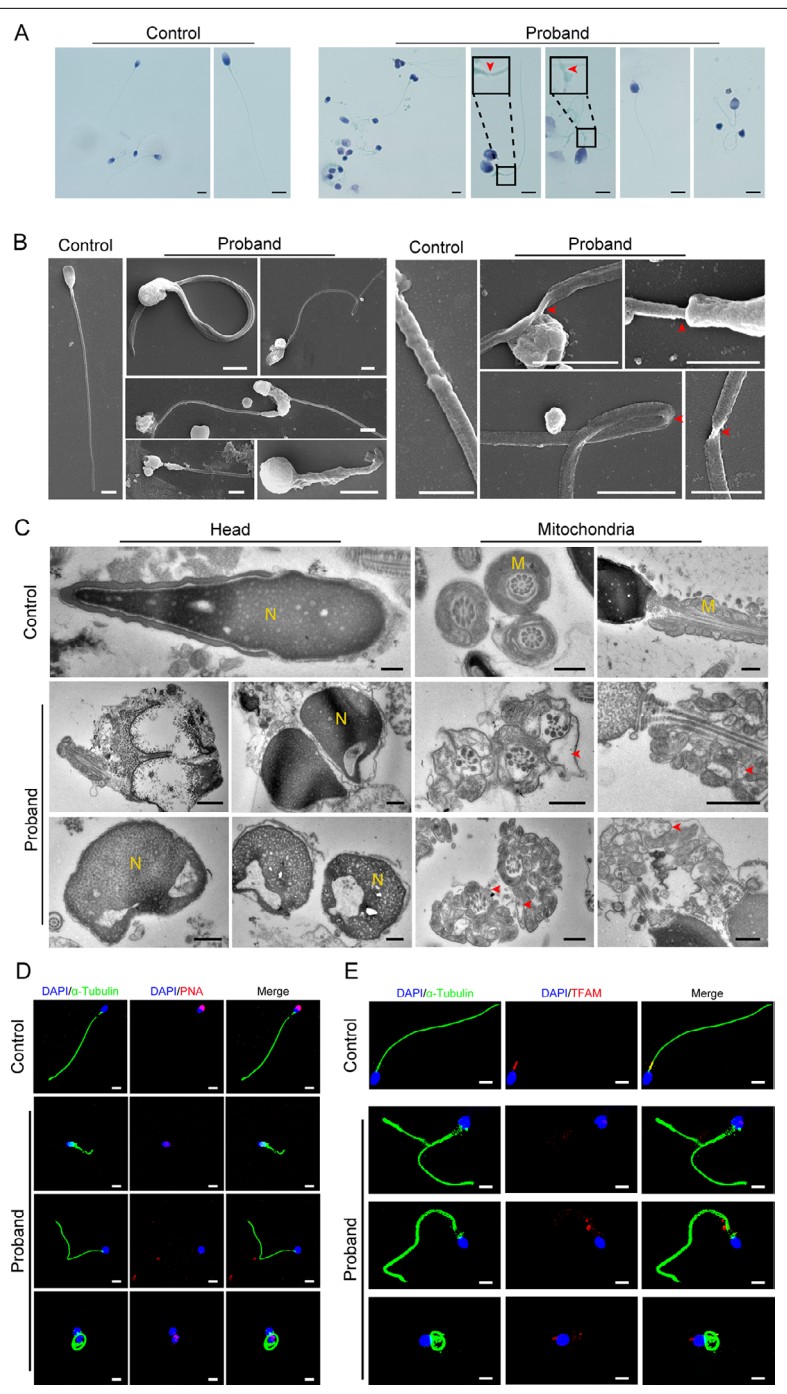

**Figure 2.** Abnormal flagellar morphology and defective acrosomes and mitochondria in the infertile patient. (**A and B**) Papanicolaou staining (**A**) and scanning electron microscopy (SEM) (**B**) results show flagellar morphological abnormalities (scale bars in A, 5 μm; scale bars in B, 2.5 μm). The dotted boxes and red arrowheads denote axoneme cracking and exposure. (**C**) Transmission electron microscopy (TEM) results showing deformed acrosomes and abnormal arrangement and diameter of mitochondria (scale bars, 500 nm). The red arrowheads denote abnormal mitochondria. N, nucleus; M, mitochondria. (**D and E**) Defects in the acrosome and mitochondria were observed in the proband's sperm by PNA (**D**) and TFAM (**E**) staining (blue, DAPI; green, α-Tubulin; red, PNA or TFAM; scale bars, 5 μm). PNA, peanut agglutinin; TFAM, transcription factor A, mitochondrial.

The online version of this article includes the following figure supplement(s) for figure 2:

**Figure supplement 1.** Transmission electron microscopy (TEM) and immunofluorescence analysis of the abnormal phenotype of spermatozoa in the patient.

**Table 2.** Semen analysis using computer-assisted sperm analysis (CASA) in the mouse model of IQ motif-containing H (*Iqch*) knockout (KO).

| | Adult Male Mice | | |
|---|---|---|---|
| | WT | KO | p[†] value |
| **Semen parameters** | | | |
| Sperm concentration ($10^6$ /ml) * | 94.51±10.68 | 67.75±3.70 | 0.038 |
| Motility (%) | 57.08±2.18 | 5.49±3.25 | <0.001 |
| Progressive motility (%) | 57.00±2.05 | 5.49±3.25 | <0.001 |
| **Sperm locomotion parameters** | | | |
| Curvilinear velocity (VCL) (µm/s) | 78.29±6.23 | 9.94±4.03 | <0.001 |
| Straight-line velocity (VSL) (µm/s) | 36.18±2.09 | 2.90±1.98 | <0.001 |
| Average path velocity (VAP) (µm/s) | 45.10±0. 19 | 4.17±2.49 | <0.001 |
| Amplitude of lateral head displacement (ALH) (µm) | 0.77±0. 02 | 0.13±0.03 | <0.001 |
| Linearity (LIN) | 0.46±0.01 | 0.27±0.08 | 0.047 |
| Wobble (WOB,=VAP/VCL) | 0.58±0.04 | 0.40±0.08 | 0.033 |
| Straightness (STR,=VSL/VAP) | 0.80±0.05 | 0.67±0.07 | 0.069 |
| Beat-cross frequency (BCF) (Hz) | 4.33±0.19 | 0.65±0.26 | <0.001 |

n = 3 biologically independent wild-type (WT) mice or knockout (KO) mice.

*Epididymides and vas deferens.

[†]two-tailed Student's t-test.

of the *Iqch* KO mice showed dilated intermembrane spaces of mitochondria and even loss of some mitochondrial material (*Figure 3C*). However, the annulus did not significantly differ between the *Iqch* KO and WT mice (*Figure 3C*), indicating that the disruptive link between the flagellar middle piece and the principal piece might result from mitochondrial defects rather than the annulus. Immunofluorescence staining of mitochondrial markers (Solute carrier family 25 member 4: SLC25A4) and SEPT4 also revealed that the mitochondria of the spermatozoa of the *Iqch* KO mice were severely defective, and there was no significant difference in the annulus of the spermatozoa between the *Iqch* KO mice and WT mice, which was consistent with the findings of the proband (*Figure 3—figure supplement 5*).

## Poor in vitro fertilization (IVF) outcomes in *Iqch* KO male mice

To evaluate the cause of the impaired fertility of the *Iqch* KO male mice, mature oocytes from WT female mice in their reproductive period were retrieved and placed into culture dishes with sperm from the *Iqch* KO mice and WT mice for IVF treatment. We found that 70% of the embryos had pronuclei in the WT group, while the fertilization rate of the *Iqch* KO group was dramatically reduced, accounting for 47% (*Figure 4A*). Consequently, the percentages of both two-cell embryos and blastocysts were significantly lower in the *Iqch* KO male mice than in the WT male mice (*Figure 4A*). Noticeably, an inactive acrosome reaction was observed in most of the sperm from the *Iqch* KO mice (*Figure 4B*), leading to the inability to cross the zona pellucida (*Figure 4—figure supplement 1*). Furthermore, PNA staining revealed abnormal acrosome development in the different development germ cells of the *Iqch* KO mice (*Figure 4—figure supplement 1*). Similarly, the expression of PNA was abnormal in the mature sperm of *Iqch* KO male mice (*Figure 4C*). In the sperm of the *Iqch* KO male mice and the proband, the PLC ζ fluorescence signal was attenuated and abnormally localized (*Figure 4D*; *Figure 4—figure supplement 1*). Taken together, these data suggested that *Iqch* might play an important role in acrosomal formation, which is essential for fertilization.

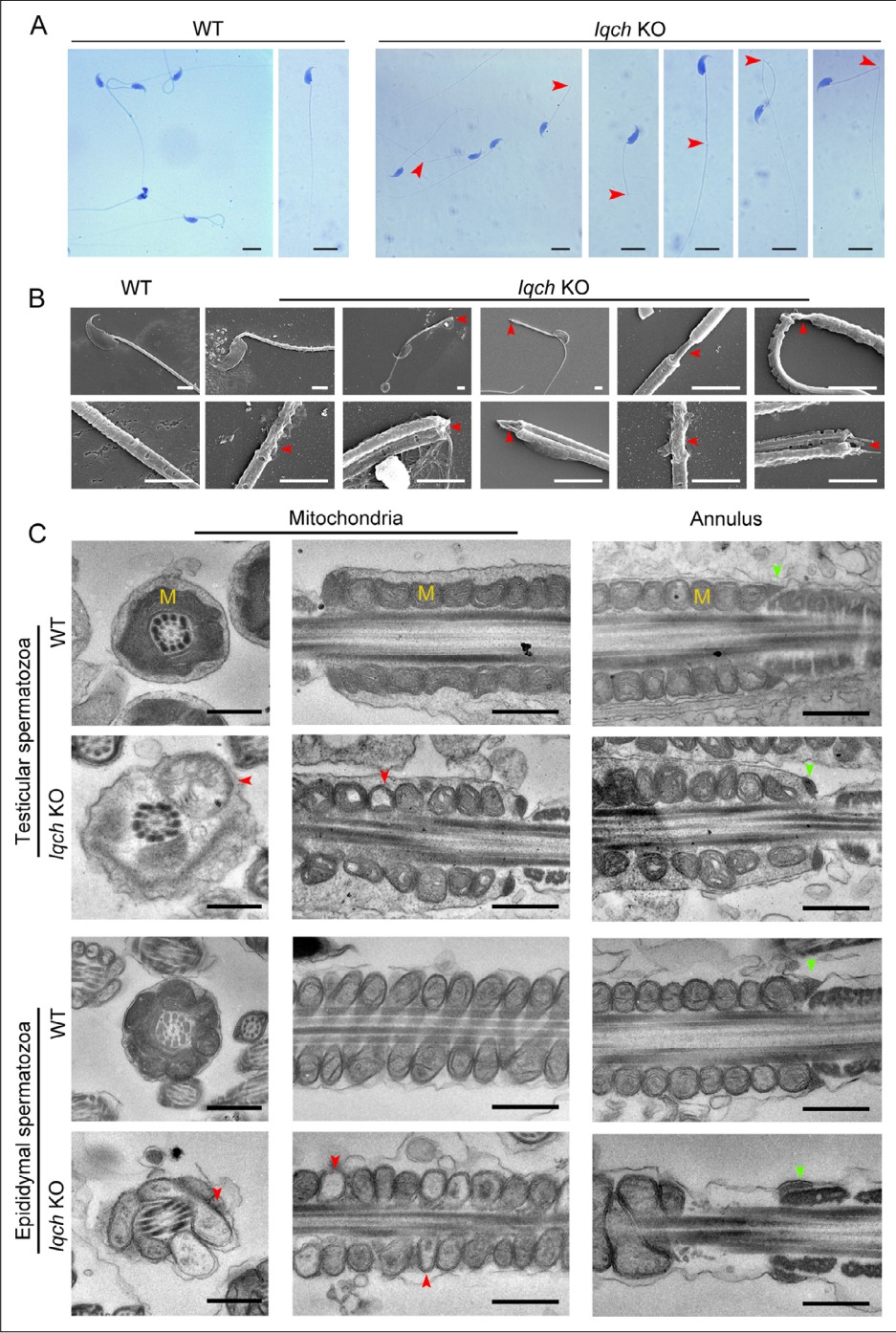

**Figure 3.** The absence of IQ motif-containing H (*Iqch*) impaired spermatogenesis in mice. (**A and B**) Papanicolaou staining (**B**) and scanning electron microscopy (SEM) (**C**) results showing unmasking, bending, or cracking of the axoneme in spermatozoa from *Iqch* knockout (KO) mice (n=3 biologically independent wild-type (WT) mice and KO mice; scale bars in B, 5 μm; scale bars in C, 2.5 μm). The red arrowheads point to axoneme abnormalities. (**C**) Transmission electron microscopy (TEM) revealing dilated intermembrane spaces of mitochondria and a normal annulus in the testicular and epididymal spermatozoa of *Iqch* KO mice (n=3 biologically independent WT mice and KO mice; scale bars, 500 nm). The red arrowheads point to the dilated intermembrane spaces of mitochondria. The green arrowheads point to the normal annulus. M, mitochondria.

The online version of this article includes the following video, source data, and figure supplement(s) for figure 3:

**Source data 1.** Primers for RT-PCR and sgRNA sequences.

*Figure 3 continued on next page*

*Figure 3 continued*

**Figure supplement 1.** The expression pattern of IQ motif-containing H (IQCH) in testes.

**Figure supplement 1—source data 1.** Original blots and gels of *Figure 3—figure supplement 1*.

**Figure supplement 2.** Generation of IQ motif-containing H (*Iqch*) knockout (KO) mice.

**Figure supplement 2—source data 1.** Original blots and gels of *Figure 3—figure supplement 2*.

**Figure supplement 3.** No obvious abnormalities in the histology of the testes or epididymis were observed in the IQ motif-containing H (*Iqch*) knockout (KO) mice.

**Figure supplement 4.** Cracking of the axoneme occurred during spermatogenesis in the IQ motif-containing H (*Iqch*) knockout (KO) mice.

**Figure supplement 5.** The spermatozoa from the IQ motif-containing H (*Iqch*) knockout (KO) mice exhibited defective mitochondria.

**Figure 3—video 1.** The movie showed the in vitro motility of epididymal sperm from wild-type (WT) mice. https://elifesciences.org/articles/88905/figures#fig3video1

**Figure 3—video 2.** Sperm with reduced motility were collected from the cauda epididymis and vas deferens of *IQ motif-containing H (Iqch)* knockout (KO) male mice and analyzed under a phase-contrast microscope(n=3 biologically independent KO mice). https://elifesciences.org/articles/88905/figures#fig3video2

## RNA-binding proteins are the most relevant targets by which IQCH regulates spermatogenesis

To elucidate the molecular mechanism by which IQCH regulates male fertility, we performed liquid chromatography-tandem mass spectrometry (LC-MS/MS) analysis using mouse sperm lysates and detected 288 interactors of IQCH (*Figure 5—source data 1*). Gene Ontology (GO) analysis of the IQCH-bound proteins revealed many particularly enriched pathways related to fertilization, sperm axoneme assembly, mitochondrion organization, calcium channel, and RNA processing (*Figure 5A*). Intriguingly, 33 ribosomal proteins were identified (*Figure 5B*), indicating that IQCH might be involved in protein synthesis. Using proteomic analysis of sperm from *Iqch* KO mice, we further assessed key proteins that might be upregulated by IQCH. A total of 1993 differentially expressed proteins were quantified, including 807 upregulated proteins and 1186 downregulated proteins (*Figure 5C*). GO analysis revealed that the significantly downregulated proteins were enriched in RNA processing, gene expression, mitochondrion biogenesis, and calcium ion regulation (*Figure 5D*), which was consistent with the enrichment of the IQCH-bound proteins.

Importantly, cross-analysis revealed that 76 proteins were shared between the IQCH-bound proteins and the downregulated proteins in *Iqch* KO mice (*Figure 5E*), suggesting that IQCH might regulate their expression by the interaction. Among the 76 proteins, 21 were RNA-binding proteins (RBPs), 10 of which were suggested to be involved in spermatogenesis (*Kuroda et al., 2000*; *Yang et al., 2012*; *Chapman et al., 2013*; *Fukuda et al., 2013*; *Legrand et al., 2019*; *Sechi et al., 2019*; *Liang et al., 2021*; *Tian and Petkov, 2021*; *Wang et al., 2023*; *Xu et al., 2022*); eight were involved in mitochondrial function; and four were calcium channel activity-related proteins (*Figure 5F*). We focused on SYNCRIP, HNRNPK, FUS, EWSR1, ANXA7, SLC25A4, and HNRPAB, which play important roles in spermatogenesis. Specifically, SYNCRIP, HNRNPK, FUS, EWSR1, and HNRPAB are RBPs that are linked to spermatogenesis by controlling mRNA translation or spermatid postmeiotic transcription (*Kuroda et al., 2000*; *Fukuda et al., 2013*; *Sechi et al., 2019*; *Tian and Petkov, 2021*; *Xu et al., 2022*). ANXA7 is a calcium-dependent phospholipid-binding protein that is a negative regulator of mitochondrial apoptosis (*Du et al., 2015*). Loss of SLC25A4 results in mitochondrial energy metabolism defects in mice (*Graham et al., 1997*). We further confirmed the binding of IQCH to these proteins by co-immunoprecipitation (co-IP) (*Figure 6A*) and confirmed the downregulation of IQCH in the sperm of *Iqch* KO mice by immunofluorescence staining and western blotting (*Figure 6B*; *Figure 6—figure supplement 1* and *Figure 6—figure supplement 2*).

Among these interactors of IQCH, HNRPAB was the most significantly downregulated protein according to proteomic analysis (*Figure 5—source data 2*), implying that HNRPAB might be the main target of IQCH. As an RBP, HNRPAB has been suggested to play an important role in spermatogenesis by regulating the translation of testicular mRNAs (*Fukuda et al., 2013*). We thus employed RNA-seq

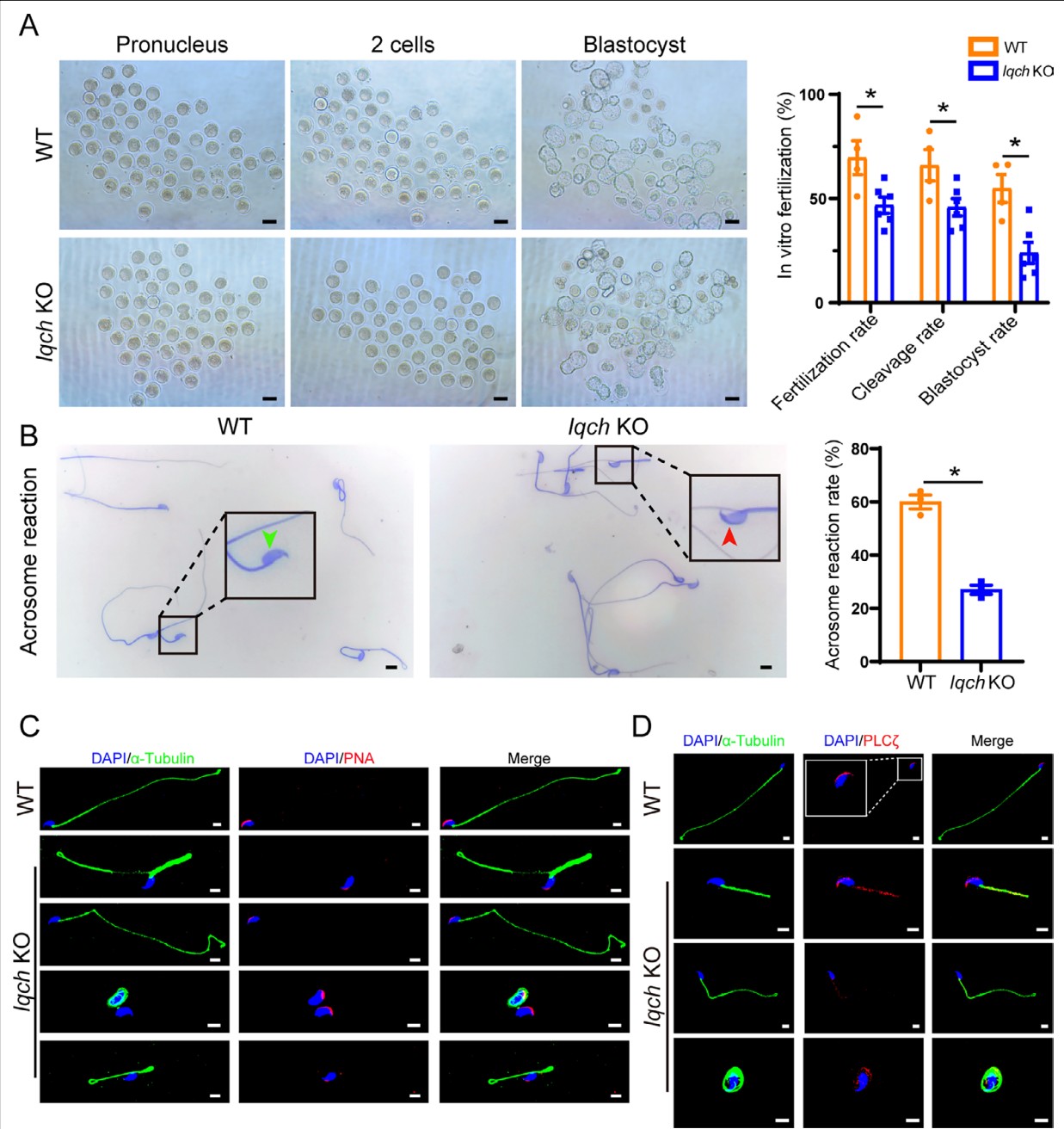

**Figure 4.** Poor in vitro fertilization (IVF) treatment outcomes resulting from the use of sperm from IQ motif-containing H (*Iqch*) knockout (KO) mice.
(**A**) Representative pronucleus embryos, two-cell embryos, and blastocysts were obtained from wild-type (WT) mice and *Iqch* KO mice. *Iqch* KO mice exhibited significantly lower fertilization rates, cleavage rates, and blastocyst formation rates than WT mice (n=3 biologically independent WT mice and KO mice; scale bars, 100 μm; Student's t-test; *p<0.05; error bars, SEM). (**B**) The acrosome reaction rates in the capacitated spermatozoa from the WT mice and *Iqch* KO mice were determined by Coomassie brilliant blue staining. The acrosome reaction rates were reduced in the spermatozoa from the *Iqch* KO mice (n=3 biologically independent WT mice and KO mice; scale bars, 5 μm; Student's t-test; *p<0.05; error bars, SEM). The green arrowheads indicate the reacted acrosomes. The red arrowheads indicate intact acrosomes. (**C and D**) PNA (**C**) and PLC ζ (**D**) staining showing abnormal acrosome morphology and aberrant PLC ζ localization and expression in spermatozoa from *Iqch* KO mice (n=3 biologically independent WT mice and KO mice; scale bars, scale bars, 5 μm). The dotted box indicates the typical pattern of PLC ζ localization and expression in spermatozoa from WT mice. PNA, peanut agglutinin; PLC ζ, phospholipase C zeta 1.

The online version of this article includes the following figure supplement(s) for figure 4:

**Figure supplement 1.** The loss of function of IQ motif-containing H (IQCH) resulted in deficiencies in oocyte activation and acrosomes.

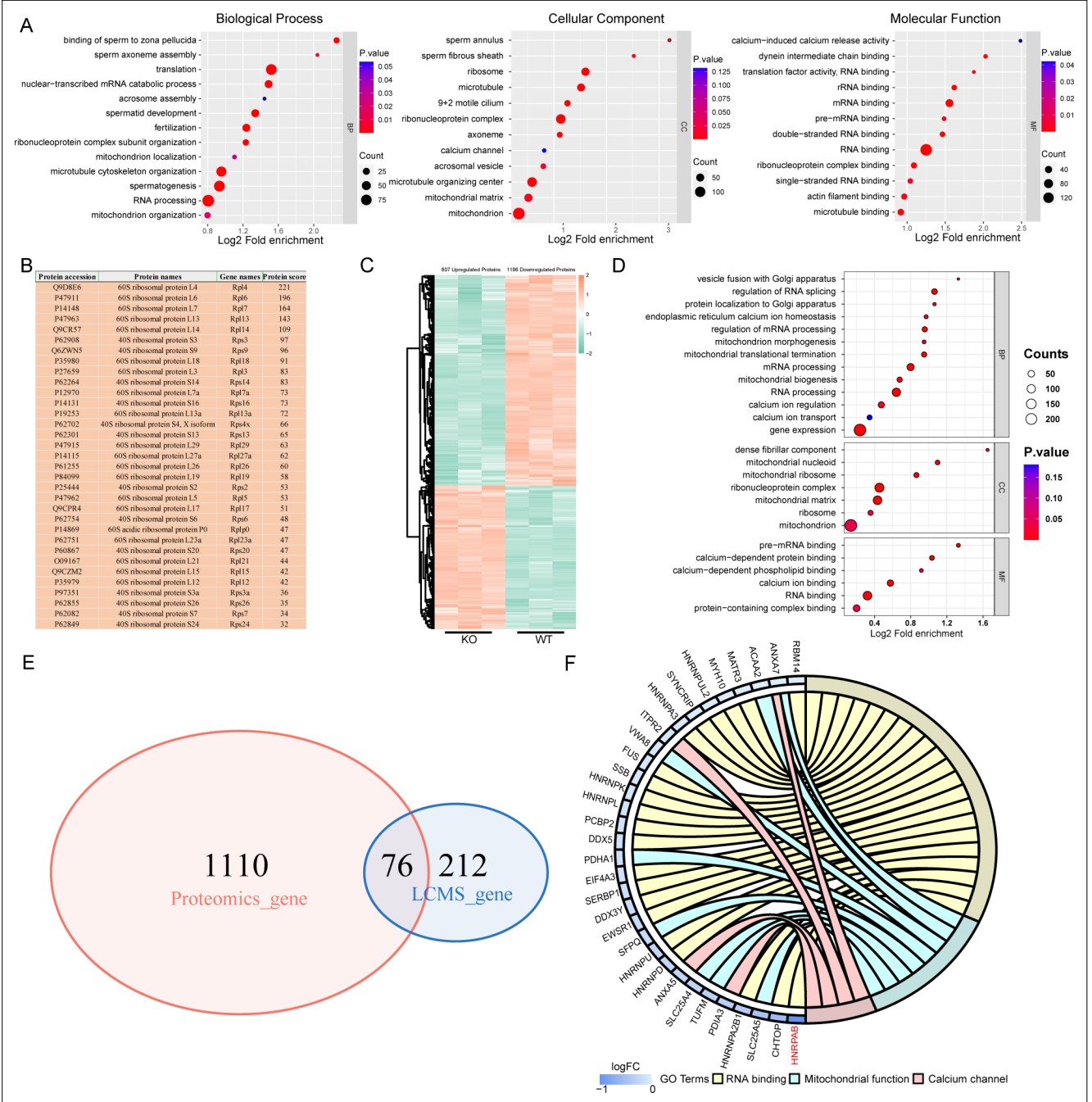

**Figure 5.** IQ motif-containing H (IQCH) bound and upregulated male reproduction-related proteins in mouse sperm. (**A**) Bubble plots of the GO analysis show that the IQCH-interacting proteins are significantly enriched in spermatogenesis and RNA processing in three categories: biological process, cellular component, and molecular function. GO, Gene Ontology. (**B**) Thirty-three ribosomal proteins interacted with IQCH in mouse sperm. (**C**) A heatmap showing the differential protein results from the proteomic analysis of the sperm from the wild-type (WT) and *Iqch* KO mice. (**D**) Bubble plots showing the decreased enrichment of proteins related to the spermatogenetic process and RNA processing according to the GO analysis of the *Iqch* KO mice compared to the WT mice. (**E**) Venn diagram depicting the 76 overlapping proteins among the 1186 downregulated proteins in sperm from *Iqch* KO mice and the 288 proteins that bind to IQCH. (**F**) The Chord diagram shows 21 proteins involved in RNA binding, eight proteins involved in mitochondrial function, and four proteins involved in calcium channel activity among the 76 overlapping proteins. HNRPAB showed the greatest reduction in expression in the *Iqch* KO mice.

The online version of this article includes the following source data for figure 5:

**Source data 1.** List of 288 interactors of IQ motif-containing H (IQCH) about LC-MS/MS analysis of mouse sperm lysates.

**Source data 2.** List of differentially down-regulated proteins about proteomic analysis of sperm from IQ motif-containing H (*Iqch*) knockout (KO) and wild-type (WT) mice.

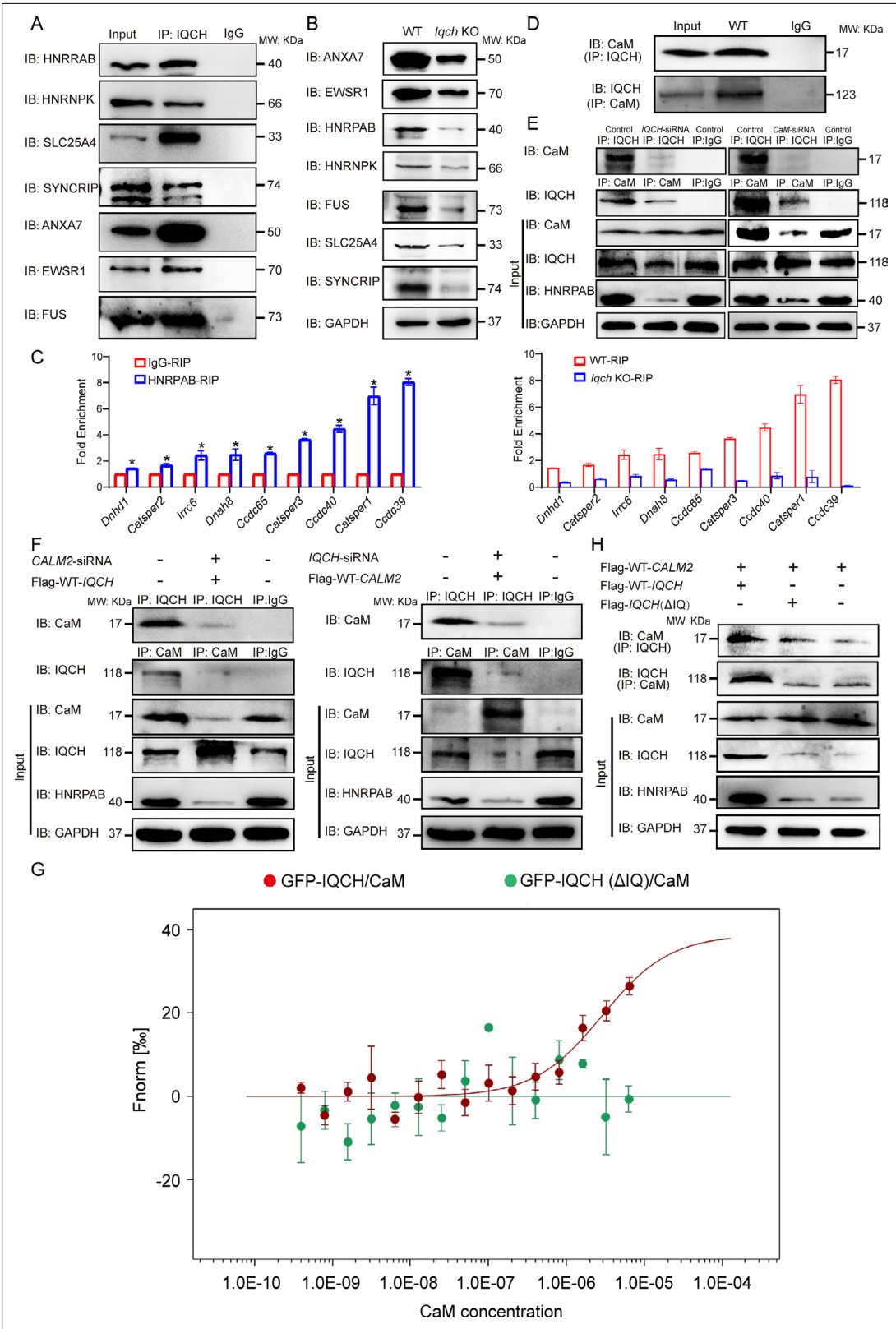

**Figure 6.** IQ motif-containing H (IQCH) interacted with calmodulin (CaM) to regulate HNRPAB expression and spermatogenesis. (**A**) Co-immunoprecipitation (co-IP) of mouse sperm lysates revealed that the seven proteins most relevant to the phenotype of the *Iqch* knockout (KO) mice were associated with IQCH. (**B**) Western blotting shows the reduced expression of the seven proteins in the sperm from the *Iqch* KO mice compared to the wild-type (WT) mice. (**C**) qPCR analysis of RNA immunoprecipitation (RIP) using HNRPAB antibodies and IgG antibodies on mouse sperm lysates

*Figure 6 continued on next page*

*Figure 6 continued*

showed that HNRPAB interacted with several RNAs associated with fertilization and axoneme assembly. qPCR analysis of RIP in the sperm lysates from the *Iqch* KO mice revealed a decrease in the interaction between HNRPAB and the RNA targets compared to that in the WT mice (Student's t-test; *p<0.05; error bars, SEM). (**D**) co-IP assays showing the binding of IQCH and CaM in WT sperm. (**E**) co-IP assays showed that the decreased expression of HNRPAB was due to the reduced binding of IQCH to CaM resulting from the knockdown of IQCH or CaM. (**F**) The overexpression of IQCH and the simultaneous knockdown of CaM or the overexpression of CaM and the simultaneous knockdown of IQCH in K562 cells further confirmed that the downregulation of HNRPAB was due to the diminished interaction between IQCH and CaM, as determined by western blotting analysis. (**G**) Analysis of the binding affinity between GFP-IQCH from the cell lysates (target) and recombinant CaM (ligand) by a microscale thermophoresis (MST) assay showing the interaction between IQCH and CaM. Their interaction was disrupted after the deletion of the IQ motif within IQCH. (**H**) co-IP of HEK293 cells cotransfected with the WT-*IQCH* and WT-*CALM2* plasmids, cotransfected with the *IQCH* (ΔIQ) and WT-*CALM2* plasmids, or cotransfected with the control and WT-*CALM2* plasmids showed that CaM interacted with the IQ motif. The downregulation of HNRPAB was due to the disrupted interaction between IQCH and CaM. Three independent experiments were performed.

The online version of this article includes the following source data and figure supplement(s) for figure 6:

**Source data 1.** Primers for qPCR and siRNA sequences.

**Source data 2.** Overview of the antibodies or dyes used in this study.

**Source data 3.** Original blots of *Figure 6*.

**Figure supplement 1.** The reduced expression levels of the seven proteins were most relevant to the phenotype of the spermatozoa of the IQ motif-containing H (*Iqch*) knockout (KO) mice.

**Figure supplement 2.** Statistics results about protein expression levels by western blotting analysis.

**Figure supplement 3.** Gene Ontology (GO) analysis of the downregulated genes of the IQ motif-containing H (*Iqch*) knockout (KO) mice through RNA sequencing.

**Figure supplement 4.** IQ motif-containing H (IQCH) interacted with calmodulin (CaM) to regulate spermatogenesis.

analysis of sperm from *Iqch* KO and WT mice to investigate the effects on RNA levels in the absence of IQCH. Importantly, among the downregulated genes, most were related to male fertility, such as axoneme assembly, spermatid differentiation, flagellated sperm motility, and fertilization (*Figure 6—figure supplement 3*). We hypothesized that this downregulation is linked to HNRPAB binding. Considering the disrupted fertilization and axoneme assembly in *Igch* KO mice, we selected the essential molecules involved in these two processes, *Catsper1*, *Catsper2*, *Catsper3*, *Ccdc40*, *Ccdc39*, *Ccdc65*, *Dnah8*, *Irrc6*, and *Dnhd1*, to confirm our speculation. We carried out RNA immunoprecipitation on formaldehyde-crosslinked sperm followed by qPCR to evaluate the interactions between HNRPAB and *Catsper1*, *Catsper2*, *Catsper3*, *Ccdc40*, *Ccdc39*, *Ccdc65*, *Dnah8*, *Irrc6*, and *Dnhd1* in KO and WT mice. As expected, binding between HNRPAB and important molecules was detected in the WT mice (*Figure 6C*), supporting the functional role of HNRPAB in testicular mRNAs. Intriguingly, significantly decreased interactions between HNRPAB and those molecules were observed in the KO mice (*Figure 6C*). Therefore, IQCH is involved in spermatogenesis mainly by regulating RBPs, especially HNRPAB, to further mediate essential mRNA expression in the testis.

## The interaction of IQCH with CaM is a prerequisite for IQCH function

Given that IQCH is a calmodulin-binding protein, we hypothesized that IQCH regulates these key molecules by interacting with CaM (*Figure 6—figure supplement 4*). As expected, CaM interacted with IQCH, as indicated by LC-MS/MS analysis. We initially confirmed the binding of IQCH and CaM in WT sperm by co-IP (*Figure 6D*). In addition, their colocalization was detected during spermatogenesis by immunofluorescence staining (*Figure 6—figure supplement 4*).

We further confirmed the binding of IQCH to CaM in K562 cells, which express both IQCH and CaM (*Figure 6E*). Moreover, a decrease in HNRPAB expression was observed when IQCH or CaM was knocked down (*Figure 6E*; *Figure 6—figure supplement 2* and *Figure 6—figure supplement 4*), suggesting that IQCH or CaM might regulate HNRPAB expression. We then overexpressed IQCH and knocked down CaM or overexpressed CaM and knocked down IQCH in K562 cells and found that the reduced HNRPAB expression could not be rescued in either of these two situations (*Figure 6F*; *Figure 6—figure supplement 2* and *Figure 6—figure supplement 4*), indicating that CaM or IQCH alone cannot mediate HNRPAB expression. These findings suggested that HNRPAB expression might be regulated by the interaction of IQCH with CaM.

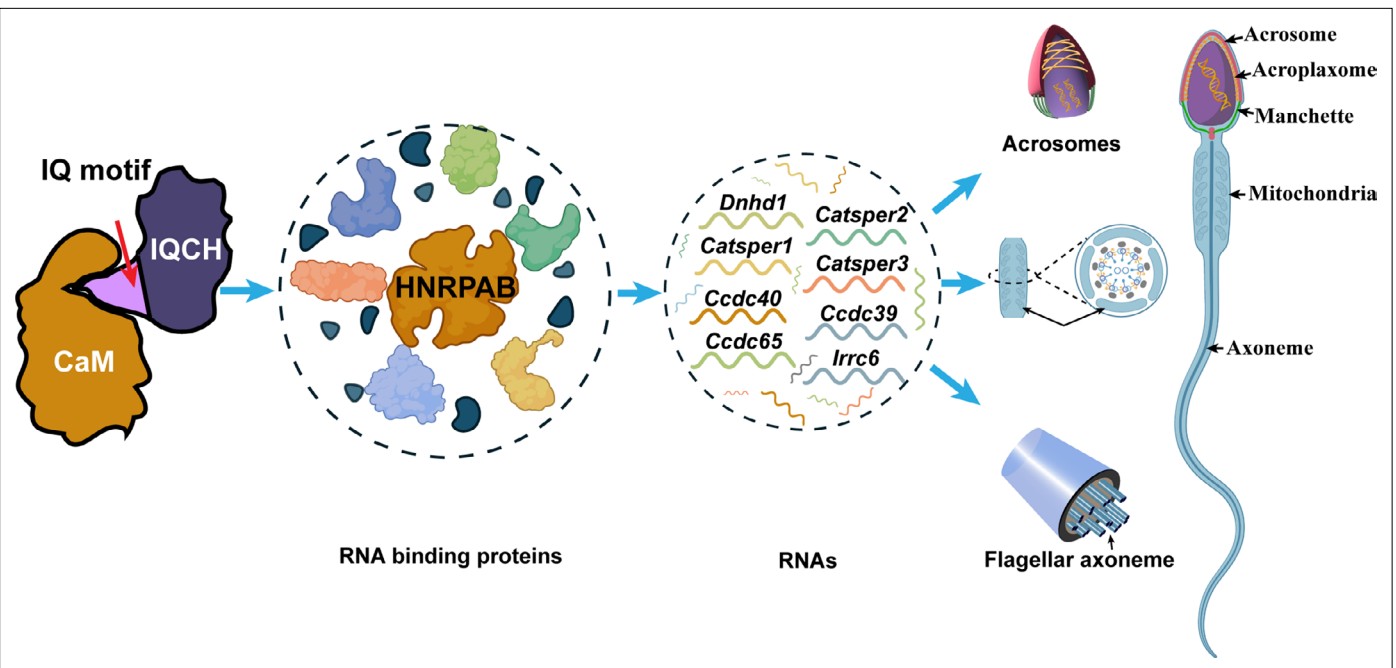

**Figure 7.** Proposed model for the mechanisms underlying the involvement of IQ motif-containing H (IQCH) in spermatogenesis. IQCH interacts with calmodulin (CaM) via the IQ motif to regulate the expression of RNA-binding proteins. RNA binding proteins, particularly HNRPAB, bind and regulate several RNAs that influence the development of the acrosome, mitochondria, and axoneme, thereby playing a critical role in spermatogenesis.

Using a microscale thermophoresis (MST) assay, we further confirmed that the IQ motif of IQCH is required for CaM binding. The binding affinity between IQCH (target) from cell lysates overexpressing GFP-*IQCH* plasmids and recombinant CaM (ligand) increased with increasing concentrations of recombinant CaM (*Figure 6G*). However, their binding was disrupted when the IQ motif of IQCH was deleted (*Figure 6G*). co-IP assays verified the above findings (*Figure 6H*; *Figure 6—figure supplement 2*). Not surprisingly, the expression of HNRPAB in cells cotransfected with the *IQCH* (△IQ)/*CALM2* plasmid and the control/*CALM2* plasmid was similar but was lower than that in cells overexpressing the WT-*IQCH* and *CALM2* plasmids (*Figure 6H*; *Figure 6—figure supplement 2*). Collectively, our findings suggest that IQCH interacts with CaM via the IQ motif to manipulate the expression of important molecules, especially HNRPAB, to play a role in spermatogenesis (*Figure 7*).

## Discussion

Throughout the evolution of divergent species, IQCH has been conserved, indicating its fundamental role in organisms. As a novel CaM-binding protein, IQCH was first identified in human and mouse testes and exclusively localized in spermatocytes and spermatids, suggesting its potential activity in spermatogenesis (*Yin et al., 2005*). However, since IQCH was identified, no other findings have supported its function in male reproduction. In this study, we found that the *IQCH* mutation impaired male fertility, as indicated by flagellar morphological abnormalities and axoneme cracking. An aberrant ultrastructure in sperm with the *IQCH* mutation was associated with severe defects in acrosomes and mitochondria. Furthermore, *Iqch* KO mice exhibited similar irregularities in the flagellum, especially the axoneme and mitochondria. We also found that *Iqch* KO mice exhibited different degrees of functional defects in acrosomes. Thus, our work supported the vital role of IQCH in flagellar and acrosome development.

To date, only four IQ motif-containing proteins, IQCD, IQCF1, IQCG, and IQCN, have been suggested to participate in spermatogenesis (*Harris et al., 2014*; *Fang et al., 2015*; *Zhang et al., 2019*; *Dai et al., 2022*). Most IQ motif-containing proteins function in Ca²⁺-dependent biological processes by binding to CaM (*Chen et al., 2014*; *Harris et al., 2014*; *Fang et al., 2015*); thus, IQCD, IQCF1, and IQCN are involved in fertilization and are relevant to the acrosome reaction, sperm capacitation, or manchette assembly (*Fang et al., 2015*; *Zhang et al., 2019*; *Dai et al., 2022*). In

addition, because CaM has an impact on the actin cytoskeleton, *Iqcg* KO mice exhibit detachment of sperm heads from tails (*Harris et al., 2014*). In our study, the *Iqch* KO mice also showed an impaired acrosome reaction, which caused reduced fertilization. Intriguingly, axoneme breaks at the annulus and mitochondrial defects were prominent in the flagella of the *Iqch* KO mice, whose phenotypes were unexplored in the previously discovered IQ motif-containing proteins. Because CaM activation can stimulate changes in the actin cytoskeleton, it is reasonable that flagellum formation is defective when IQCH is absent. $Ca^{2+}$ is a key player in the regulation of mitochondrial functions (*Bravo-Sagua et al., 2017*). Thus, the absence of IQCH leads to a failure of interaction with CaM and disruption of normal $Ca^{2+}$ signaling, which consequently causes mitochondrial defects in *Iqch* KO mice and patients with *IQCH* variants. Our findings suggested that fertilization is the main action of IQ motif-containing proteins, and each specific IQ motif-containing protein also has its own distinct role in spermatogenesis.

However, there are few publications regarding the underlying mechanism by which IQ motif-containing proteins manipulate reproductive processes, except for a recent study showing that IQCN interacts with CaM to regulate the expression of CaM-binding proteins, including manchette-related proteins (LZTFL1, KIF27, and RSPH6A), IFT family proteins and their motor proteins (IFT22, IFT43, IFT74, IFT81, IFT140, IFT172, WDR19, TTC21B, and DYNC2H1), and ribosomal protein family proteins (RPS5, RPS25, RPS27, and RPSA) (*Dai et al., 2022*). In our study, IQCH was shown to regulate the expression of RBPs via interactions with these proteins. RBPs are a large class of proteins that assemble with RNAs to form ribonucleoproteins (RNPs). RBPs function at various stages of RNA processing, including alternative splicing, RNA modification, nuclear export, localization, stability, and translation efficiency (*Corley et al., 2020*). RNA processing in male reproductive biology is essential for the production of mature sperm, and many RBPs have been demonstrated to be indispensable during spermatogenesis (*Morgan et al., 2021*). Among the RBPs, HNRPAB, a heterogeneous nuclear RNP, showed the most significant reduction among the downregulated interactors of IQCH, suggesting that HNRPAB might be the most important downstream effector protein of IQCH. It has been reported that HNRPAB plays a central role in spermatogenesis by regulating stage-specific translation of testicular mRNAs (*Fukuda et al., 2013*). As expected, an RNA-binding protein immuno-precipitation (RIP) assay using an anti-HNRPAB antibody revealed binding between HNRPAB and the mRNAs of several important genes (*Catsper1*, *Catsper2*, *Catsper3*, *Ccdc40*, *Ccdc39*, *Ccdc65*, *Dnah8*, *Irrc6*, and *Dnhd1*) involved in spermatogenesis. We further found that the interaction between IQCH and CaM is a prerequisite for regulating HNRPAB expression. We thus hypothesized that IQCH regulates male fertility by binding to CaM and controlling HNRPAB to manipulate the expression of key mRNAs involved in spermatogenesis.

In conclusion, our study identified a novel IQ motif-containing protein-coding gene, *IQCH/Iqch*, which is responsible for spermatogenesis in humans and mice, broadening the scope of known male infertility-associated genes and thus further illustrating the genetic landscape of this disease. In addition, our study suggested an unexplored mechanism in which IQCH regulates the expression of key RBPs involved in spermatogenesis, especially HNRPAB, by interacting with CaM to play crucial roles in fertilization and axoneme assembly. We believe that our findings will provide genetic diagnostic markers and potential therapeutic targets for infertile males with *IQCH* pathogenic variants.

# Materials and methods
## Study participants
A family with male infertility was recruited from West China Second University Hospital, and healthy Chinese volunteers were enrolled as controls. The study was conducted according to the tenets of the Declaration of Helsinki, and ethical approval (No. 2019040) was obtained from the Ethical Review Board of West China Second University Hospital, Sichuan University. Each subject signed an informed consent form.

## WES and Sanger sequencing
Peripheral blood samples were obtained from all subjects, and genomic DNA was extracted using a QIAamp DNA Blood Mini Kit (QIAGEN, 51126). The exomes of the subjects were captured by Agilent SureSelect Human All Exon V6 Enrichment kits (Agilent) and then sequenced on a HiSeq X-TEN system

(Illumina, USA). All reads were mapped to the human reference sequence (UCSC Genome Browser hg19) using Burrows-Wheeler Alignment. After quality filtration with the Genome Analysis Toolkit, functional annotation was performed using ANNOVAR through a series of databases, including the 1000 Genomes Project, dbSNP, HGMD, and ExAC. PolyPhen-2, SIFT, MutationTaster, and CADD were used for functional prediction. Variant verification of *IQCH* in patients was confirmed by Sanger sequencing using the primers listed in *Figure 1—source data 1*.

## Minigene assay

A minigene splicing assay was used to explore the effect of the *IQCH* variant (c.387+1_387+10 del) on splicing. The WT-*IQCH* and Mut-*IQCH* sequences, including intron 3, exon 4, and intron 4, were PCR-amplified separately from the genomic DNA of the control and patient. The two amplified fragments were cloned and inserted into the minigene vector pSPL3 between the EcoRI and BamHI sites through the Basic Seamless Cloning and Assembly Kit (TransGen Biotech, China, CU201-02). After transfection into HEK293T cells by DNA and siRNA transfection reagent (Polypus, 101000046), the splicing patterns of the transcripts produced from the WT-*IQCH* and Mut-*IQCH* plasmids were analyzed by RT–PCR, gel electrophoresis, and Sanger sequencing. The primers used for plasmid genomic amplification and RT–PCR are listed in *Figure 1—source data 1*.

## *Iqch* knockout mice

The animal experiments were approved by the Experimental Animal Management and Ethics Committee of West China Second University Hospital, Sichuan University (No. 2021033). All animal procedures complied with the guidelines of the Animal Care and Use Committee of Sichuan University. The mouse *Iqch* gene has six transcripts and is located on chromosome 9. According to the structure of the *Iqch* gene, exons 2 ~3 of the *Iqch-201* (ENSMUST00000042322.10) transcript were recommended as the knockout region. The region contains a 215 bp coding sequence, and deletion of this region was expected to disrupt IQCH protein function. The sgRNA sequences synthesized by Sangon Biotech are listed in *Figure 3—source data 1*. The two complementary DNA oligos of each sgRNA target were annealed and ligated to the pUC57-sgRNA plasmid (Addgene) for cloning. The recombinant plasmid was transformed into DH5α competent cells, and positive clones were screened based on kanamycin resistance and sequencing. The recombinant plasmid was linearized and purified by phenol-chloroform extraction. In vitro transcription of the sgRNAs was performed using the MEGAshortscript Kit (Ambion, AM1354), and the sgRNAs were purified using the MEGAclear Kit (Ambion, AM1908). Cas9 mRNA was purchased from TriLink BioTechnologies. The mouse zygotes were coinjected with an RNA mixture of Cas9 mRNA (~50 ng/μl) and sgRNA (~30 ng/μl). The injected zygotes were transferred into pseudopregnant recipients to obtain the F0 generation. DNA was extracted from the tail tissues of the 7-day-old offspring, and PCR amplification was carried out with genotyping primers. A stable F1 generation (heterozygous mice) was obtained by mating the positive F0 generation mice with wild-type (WT) C57BL/6JG-pt mice. The primers used for genotyping are listed in *Figure 3—source data 1*.

## Mouse fertility testing

To confirm the fertility of the *Iqch* KO male mice, natural mating tests were conducted. Briefly, six *Iqch* KO and six littermate WT sexually mature male mice (8–12 weeks old) were paired with two 6- to 8-week-old normal C57BL/6 J females (each male was mated with two female mice) for 6 months. Regarding the fertility of *Iqch* KO female mice, WT (n=15) and *Iqch* KO (n=9) females were mated with WT males for 6 months. The presence of vaginal plugs was used as an indicator of successful mating and reproductive behavior. The vaginal plugs of the mice were examined every morning. Female mice with vaginal plugs were fed separately, and the number of pups per litter was recorded on the day of birth.

## IVF

Eight-week-old C57BL/6 J female mice were superovulated by injecting 5 IU of pregnant mare serum gonadotropin (PMSG) followed by 5 IU of human chorionic gonadotropin (hCG) 48 hr later. Sperm was released from the cauda epididymis of 10-week-old male mice, and sperm capacitation was performed for 50 min using test yolk buffer with HEPES (TYH) solution. Cumulus-oocyte complexes (COCs) were

obtained from the ampulla of the uterine tube 14 hr after hCG injection. The ampulla was torn with a syringe needle, and the COCs were gently squeezed onto liquid drops of human fallopian tube fluid (HTF) medium. The COCs were then incubated with ~5 µl of the sperm suspension in HTF liquid drops at 37 °C under 5% $CO_2$. After 6 hr, the eggs were washed several times using HTF medium to remove cumulus cells and then transferred to liquid drops of potassium simplex optimized medium (KSOM).

## Acrosome reaction analysis

The spermatozoa were collected from the cauda epididymis and capacitated for 50 min in TYH medium at 37 °C under 5% $CO_2$. Highly motile sperm were collected from the upper portion of the medium, and 10 µM calcium ionophore A23187 (Sigma-Aldrich, C7522) was added to induce the acrosome reaction. After 15 min, the spermatozoa were spotted on a glass microscope slide, dried, and fixed with 4% paraformaldehyde (PFA) for 10 min. The acrosomes were stained with Coomassie brilliant blue.

## H&E staining

For staining of the testicular tissues, samples were dissected from the adult mice and fixed in 4% PFA overnight at 4 °C. The fixed tissues were embedded in paraffin, sectioned (5 µm thick), dewaxed, and rehydrated. The testis sections were stained with hematoxylin and eosin solution (Beyotime, C0105M) before imaging using a microscope (Leica, Germany).

## Immunofluorescence staining

For sperm immunostaining, fresh sperm samples were fixed with 4% PFA for 20 min at room temperature. For the staining of the testis tissues, heat-induced antigen retrieval was performed in a citrate antigen retrieval solution (Beyotime, P0081) according to the manufacturer's instructions. After permeabilization with 1% Triton X-100 for 30 min, the sperm slides were blocked with 5% bovine serum albumin serum for 1 hr at room temperature. Primary antibodies were added to the slides, which were then incubated overnight at 4 °C. After washing three times with PBS, the slides were incubated with secondary antibodies for 1 hr at room temperature. The nuclei were counterstained with DAPI dye (Sigma-Aldrich, 28718-90-3) and mounted with an antifade mounting medium. Images were captured with a laser-scanning confocal microscope (Olympus, Japan). Detailed information on the antibodies is provided in *Figure 6—source data 2*.

## qPCR

Total RNA was extracted from mouse testes using an RNA Easy Fast Tissue/Cell Kit (Tiangen Biotech, China, 4992732). Approximately 0.3 mg of total RNA was converted into cDNA with the PrimeScript RT Reagent Kit (Takara, RR037A) according to the manufacturer's instructions. The cDNAs were individually diluted 10-fold for use as templates for subsequent qPCR with iTaq Universal SYBR Green Supermix (Bio-Rad, 1725124). Mouse *Gapdh* was used as an internal control. The mRNA expression of the detected targets was quantified according to the $2^{-\Delta\Delta Ct}$ method. The primers used for qPCR are listed in *Figure 3—source data 1* and *Figure 6—source data 1*.

## Papanicolaou staining

Semen samples were fixed in 4% paraformaldehyde, and the coated slides were air-dried before being rehydrated with 80%, 70%, and 50% ethanol and distilled water. The samples were then stained with Lea's hematoxylin, rinsed with distilled water, and stained with G-6 orange stain and EA-50. Following staining, the slides were dehydrated with ethanol and mounted.

## SEM

The spermatozoa were fixed in 2.5% phosphate-buffered glutaraldehyde (GA) at room temperature for 30 min and then deposited on coverslips. The coverslips were dehydrated via an ascending gradient of 50%, 70%, 95%, and 100% ethanol and air-dried. The specimens were then attached to specimen holders and coated with gold particles using an ion sputter coater before being viewed with a JSM-IT300 scanning electron microscope (JEOL, Japan).

## TEM

The precipitate of the spermatozoa was fixed with 2.5% (vol/vol) glutaraldehyde in 0.1 M phosphate buffer (PB) (pH 7.4) and washed two times in PB and two times in ddH$_2$O. Then, the tissues were immersed in 1% (wt/vol) OsO$_4$ and 1.5% (wt/vol) potassium ferricyanide aqueous solution at 4 °C for 2 hr. After washing, the samples were dehydrated through graded alcohol (30%, 50%, 70%, 80%, 90%, 100%, 10 min each) in pure acetone (10 min twice). The samples were then infiltrated in a graded mixture (3:1, 1:1, 1:3) of acetone and SPI-PON812 resin (21 ml of SPO-PON812, 13 ml of DDSA and 11 ml of NMA), and then the pure resin was changed. The specimens were embedded in pure resin with 1.5% BDMA and polymerized for 12 hr at 45 °C and 48 hr at 60 °C. Ultrathin sections (70 nm thick) were cut with a microtome (Leica EM UC6, Germany), double-stained with uranyl acetate and lead citrate, and examined by a transmission electron microscope (TECNAI G2 F20, Philips, USA).

## Proteomic quantification

Proteins were extracted from the sperm samples using radioimmunoprecipitation assay (RIPA) lysis buffer (Applygen, C1053) supplemented with 1 mM phenylmethanesulfonyl fluoride (PMSF) and protease inhibitors on ice. The supernatants were collected following centrifugation at 12,000×g for 20 min. Protein concentrations were calculated by Bradford quantification and sodium dodecyl sulfate-polyacrylamide gel electrophoresis (SDS-PAGE). An enzyme solution at a ratio of 1:20 of trypsin enzyme (μg) to substrate protein (μg) was added to 100 μg of the protein samples, which were subsequently vortexed, centrifuged at low speed for 1 min, and incubated at 37 °C for 4 hr. The peptide liquid obtained after salt removal was freeze-dried. The peptide sample was dissolved in 0.5 M TEAB, added to the corresponding iTRAQ labeling reagent, and stored at room temperature for 2 hr. The Shimadzu LC-20AB liquid phase system was used for purification, and the separation column was a 5 μm 4.6×250 mm Gemini C18 column for liquid phase separation of the sample. The dried peptide samples were reconstituted with mobile phase A (2% can, 0.1% FA) and centrifuged at 20,000×g for 10 min, after which the supernatant was collected for injection. The separation was performed by an UltiMate 3000 UHPLC (Thermo Fisher, USA). The sample was first enriched in a trap column, desalted, and then placed into a self-packed C18 column. The peptides separated by liquid phase chromatography were ionized by a nanoESI source and then passed to a Q Exactive HF X tandem mass spectrometer (Thermo Fisher, USA) for DDA (data-dependent acquisition) mode detection. The raw data were converted to mgf files for bioinformatics analysis, and protein identification from tandem mass spectra was performed by database searching (UniProt). The protein quantification process included the following steps: protein identification, tag impurity correction, data normalization, missing value imputation, protein ratio calculation, statistical analysis, and results presentation. Proteins with a 1.5-fold change and a p-value (using Student's t-test) less than 0.05 were defined as differentially expressed proteins.

## RNA sequencing

Total RNA (~1 μg) was isolated from fresh spermatozoa from the WT mice (n=3) and the *Iqch* KO mice (n=3) using the RNAsimple Total RNA Kit (Tiangen Biotech, 4992858). A Ribo-Zero rRNA Removal Kit (Ilumina, MRZPL1224) was used to remove rRNA from the samples. RNA integrity was evaluated using an Agilent 2100 Bioanalyzer system (Agilent Technologies). An mRNA sequencing library for RNA-seq was constructed using the TruSeq RNA Library Prep Kit v2 (Illumina, RS-122–2001), followed by paired-end (2×100 bp) sequencing using the Illumina HiSeq 4000 Sequencing System (Illumina, USA) at the Beijing Genomics Institute. The raw paired-end reads were filtered through a FASTX Toolkit filter. The quality of the reads was confirmed by FastQC (ver: 0.11.3). The raw data (raw reads) in the fastq format were processed through in-house Perl scripts. The feature counts (ver: l.5.0-p3) were used to count the read numbers mapped to each gene. The fragments per kilobase million (FPKM) of each gene were calculated based on the length of the gene and the read count mapped to the gene. Signaling pathway matching analysis for the list of differentially expressed genes (DEGs) was performed using the Kyoto Encyclopedia of Genes and Genomes (KEGG) Mapper (https://www.genome.jp/kegg/). The mapping of GO to DEGs was carried out using Blast2GO (ver: 4.1.9).

## LC-MS/MS analysis

Sperm from WT mice were lysed using RIPA buffer supplemented with 1 mM PMSF and protease inhibitors (Applygen, P1265) on ice. The lysates were then centrifuged at 14,000×g for 15 min at 4 °C, and the clear supernatants were incubated with 5 µg IQCH antibody (the rabbit polyclonal IQCH antibody was synthesized using the immunogen KAEAATKIQATWKSYKARSSFISYRQKKWA) at 4 °C overnight. The next day, the incubated product was mixed with 50 µL protein A/G magnetic beads (Bimake, B23202) and incubated for 2 hr at room temperature with rotation. Subsequent washes of the immunoprecipitates were performed three times using a wash buffer (50 mM Tris-HCl, 150 mM NaCl, 0.1% Triton X-100) and boiled in 1x SDS PAGE buffer at 95 °C for 10 min. The samples were separated by SDS-PAGE and subjected to Coomassie brilliant blue staining. Protein bands were then excised from the gel for subsequent LC-MS/MS analysis. Subsequent LC-MS/MS analysis was performed using a Q Exactive HF-X mass spectrometer (Thermo Fisher, USA) equipped with a nanoelectrospray ion source at Jingjie PTM BioLabs. The raw data files were queried against the Mus_musculus_10090_SP_20210721.fasta database, which contains 17,089 entries. The search employed MaxQuant (v.1.6.15.0) and its integrated Andromeda search engine, utilizing default settings, to identify proteins and ascertain their respective label-free quantification (LFQ) values. The LC-MS/MS results underwent two main processing steps: (i) exclusion of contaminant proteins, reverse sequences, and proteins identified solely by site and (ii) normalization of LFQ intensity. The normalization utilized the median of commonly identified proteins in the sample, with missing values imputed by the minimum value.

## RIP

RIP assays were utilized to isolate RNA binding with HNRPAB, following a previous study (*Fuentes-Iglesias et al., 2020*). Briefly, testicular tissue was disaggregated using a cryogenic tissue homogenizer and subsequently crosslinked in 1% formaldehyde (w/v) for 10 min. Crosslinking was halted by 2.5 M glycine and 25 mM Tris-base buffer for 5 min, and the samples were lysed in RIP lysis buffer (50 mM Tris-HCl pH 7.5, 150 mM NaCl, 1% NP-40, 1 mM DTT, 1 mM PMSF, and 100 U/ml RNase Inhibitor) on ice for 10 min. The lysates were then sonicated in an ice bath for five cycles (10 s on, 20 s off), followed by centrifugation at 14,000 g for 10 min at 4 °C. The supernatant (10% volume was kept as input) was incubated overnight at 4 °C with primary antibodies. Anti-HNRPAB antibody and mouse IgG were employed in the RIP assays. The lysates were subsequently incubated with 75 µL of protein A/G magnetic beads at 4 °C for 4 hr with rotation. The beads were washed three times with RIP washing buffer (20 mM Tris-HCl pH 7.5, 150 mM NaCl, 0.1% NP-40, 0.1% SDS, 100 U/ml RNase inhibitor) and then incubated at 70 °C for 45 min to reverse the crosslinks. Target RNAs from the input and eluate were extracted using TRIzol reagent (Invitrogen, 10296028CN) and reverse-transcribed with the PrimeScript RT Reagent Kit (Takara, RR037A), according to the manufacturer's instructions.

## Western blotting and co-IP

Proteins were extracted from cultured cells and mouse sperm using a lysis buffer on ice. The supernatants were collected following centrifugation at 14,000×g for 20 min. The proteins were electrophoresed on 10% SDS-PAGE gels and transferred to nitrocellulose membranes (GE Healthcare). The blots were blocked in 5% milk and incubated with primary antibodies overnight at 4 °C, followed by incubation with anti-rabbit or anti-mouse IgG heavy and light chain (H&L) (horseradish peroxidase, HRP) (Abmart, M21002, and M21001) at a 1/10,000 dilution for 1 hr. The signals were evaluated using Super ECL Plus western blotting Substrate (Applygen, P1050) and a Tanon-5200 Multi chemiluminescence imaging system (Tanon, China).

For the co-IP assays, the extracted proteins were incubated with primary antibodies overnight at 4 °C. The lysates were then incubated with 20 µl of Pierce Protein A/G-conjugated Agarose for 2 hr at room temperature. The beads were washed with washing buffer, eluted with 1.2x SDS loading buffer, and boiled for 5 min at 95 °C. Finally, the products were separated via SDS-PAGE and analyzed by immunoblotting. Detailed information on the antibodies used in the western blotting experiments is provided in *Figure 6—source data 2*.

## Antibody production

The peptide used for raising the anti-IQCH antibody was derived from amino acid residues 406–435 (KAEAATKIQATWKSYKARSSFISYRQKKWA) of mouse IQCH. The peptide coupled with keyhole limpet

hemocyanin (KLH) (Sigma–Aldrich, H7017) was dissolved in saline, emulsified with 1 ml of Freund's complete adjuvant (Beyotime, P2036), and injected at multiple sites on the backs of New Zealand white rabbits. The antiserum was collected within 2 weeks after the final injection.

## Cell culture and transfection

We purchased K562 cells (CRL-3344) and HEK293T cells (CRL-11268) from the American Type Culture Collection. The identities of the cell lines have been authenticated by STR profiling. All cell lines were mycoplasma negative. The K562 cell line is a human leukemia cell line showing high expression levels of IQCH and CaM and was used for the knockdown experiments. K562 cells were cultured in Basic Roswell Park Memorial Institute 1640 Medium (Gibco, C11875500BT) supplemented with 10% fetal bovine serum (Gibco, 12483020). HEK293T cells were cultured in Dulbecco's modified Eagle medium (Gibco, 11965092) supplemented with 10% fetal bovine serum (Gibco, 12483020). The expression plasmids pcDNA3.1-Flag-*CALM2*, pCMV-MCS-3* flag-WT-*IQCH,* and pCMV-MCS-3* flag-*IQCH* (△IQ) and small interfering RNAs (siRNAs) targeting *CALM2* and *IQCH* were constructed by Vigene Biosciences (Jinan, China). The sequences of the siRNAs used are listed in *Figure 6—source data 1*. The plasmids and siRNAs were transfected into cells with jetPRIME transfection reagents (Polypus, 101000046) according to the manufacturer's protocol.

## MST assay

The MST experiments were conducted on a Monolith NT.115 system (NanoTemper Technologies, Germany). Lysates of the HEK293T cells transfected with fluorescent GFP-*IQCH* or GFP-*IQCH*(△IQ) were normalized by raw fluorescence (count), diluted using MST buffer (50 mM Tris-HCl pH 7.5, 150 mM NaCl, 10 mM MgCl2, 0.05% (v/v) Tween 20), and added to 16 PCR tubes (10 µl per tube). Then, the purified CaM was diluted into 16 gradients using MST buffer. Ten microliters of different concentrations of CaM were mixed with 10 µl of fluorescent GFP-IQCH protein and GFP-*IQCH*(△IQ) and reacted in a dark box for 15 min at room temperature. The samples were added to monolith capillaries (NanoTemper, MO-L022) and subsequently subjected to MST analysis. The measurement protocol times were as follows: fluorescence before 5 s, MST after 30 s, fluorescence after 5 s, and delay of 25 s. The dissociation constant (Kd) was determined using a single-site model to fit the curve.

## Statistical analysis

The data were compared for statistical significance using GraphPad Prism version 9.0.0 (GraphPad Software). The unpaired, two-tailed Student's t-test was used for the statistical analyses. The data are presented as the mean ± SEM, and statistically significant differences are represented as *p<0.05.

# Acknowledgements

We thank the patient and the family members for their support during this research study. This study was funded by the National Key Research and Development Project (2019YFA0802101)

# Additional information

### Funding

| Funder | Grant reference number | Author |
| --- | --- | --- |
| National Key Research and Development Program of China | 2019YFA0802101 | Suren Chen |

The funders had no role in study design, data collection and interpretation, or the decision to submit the work for publication.

### Author contributions

Tiechao Ruan, Data curation, Formal analysis, Writing – original draft; Ruixi Zhou, Junchen Guo, Chuan Jiang, Gan Shen, Siyu Dai, Data curation; Yihong Yang, Resources; Xiang Wang, Data curation,

Methodology; Suren Chen, Data curation, Writing – original draft; Ying Shen, Data curation, Supervision, Writing – original draft, Writing - review and editing

### Author ORCIDs
Tiechao Ruan (ID) https://orcid.org/0000-0002-2924-0230
Xiang Wang (ID) http://orcid.org/0000-0002-8930-6801
Suren Chen (ID) https://orcid.org/0000-0002-9337-5412
Ying Shen (ID) http://orcid.org/0000-0002-6346-1002

### Ethics

The study was conducted according to the tenets of the Declaration of Helsinki, and ethical approval (No. 2019040) was obtained from the Ethical Review Board of West China Second University Hospital, Sichuan University. Each subject signed an informed consent form.

The animal experiments were approved by the Experimental Animal Management and Ethics Committee of West China Second University Hospital, Sichuan University (No. 2021033).

Reviewer #3 (Public review): https://doi.org/10.7554/eLife.88905.6.sa1
Author response https://doi.org/10.7554/eLife.88905.6.sa2

---

# Additional files

### Supplementary files
• MDAR checklist

### Data availability

All data generated or analysed during this study are included in the manuscript and supporting files; Source data files have been provided for Figures 1, 3, 5, and 6.

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
