## [Editor Report · eLife assessment]

This **valuable** study describes mice with a knock out of the IQ motif-containing H (IQCH) gene, to model a human loss-of-function mutation in IQCH associated with male sterility. While the evidence for interaction between IQCH and potential RNA binding proteins is limited, the human infertility is reproduced in the mouse, making it a **compelling** model. The paper could be of interest to cell biologists and male reproductive biologists working on the sperm flagellar cytoskeleton and mitochondrial structure.

---

## [Referee Report · Reviewer #3 (Public review)]

In this study, Ruan et al. investigate the role of the IQCH gene in spermatogenesis, focusing on its interaction with calmodulin and its regulation of RNA-binding proteins. The authors examined sperm from a male infertility patient with an inherited IQCH mutation as well as Iqch CRISPR knockout mice. The authors found that both human and mouse sperm exhibited structural and morphogenetic defects in multiple structures, leading to reduced fertility in Ichq-knockout male mice. Molecular analyses such as mass spectrometry and immunoprecipitation indicated that RNA-binding proteins are likely targets of IQCH, with the authors focusing on the RNA-binding protein HNRPAB as a critical regulator of testicular mRNAs. The authors used in vitro cell culture models to demonstrate an interaction between IQCH and calmodulin, in addition to showing that this interaction via the IQ motif of IQCH is required for IQCH's function in promoting HNRPAB expression. In sum, the authors concluded that IQCH promotes male fertility by binding to calmodulin and controlling HNRPAB expression to regulate the expression of essential mRNAs for spermatogenesis. These findings provide new insight into molecular mechanisms underlying spermatogenesis and how important factors for sperm morphogenesis and function are regulated.

The strengths of the study include the use of mouse and human samples, which demonstrate a likely relevance of the mouse model to humans; the use of multiple biochemical techniques to address the molecular mechanisms involved; the development of a new CRISPR mouse model; ample controls; and clearly displayed results. Assays are done rigorously and in a quantitative manner. Overall, the claims made by the authors in this manuscript are well-supported by the data provided.

---

## [Author Response]

The following is the authors’ response to the previous reviews

It is unclear to us why you did not adjust the title to better reflect the well-supported claims of the paper, i.e., that this is a valuable model for human loss-of-function mutations in IQCH.

Thanks for the editor’s suggestion. We have changed the title to “Deficiency of IQCH causes male infertility in humans and mice.” Additionally, we have provided the original images of the gels or blots as a zipped folder.